# Methods for a Non-Targeted Qualitative Analysis and Quantification of Benzene, Toluene, and Xylenes by Gas Chromatography-Mass Spectrometry of E-Liquids and Aerosols in Commercially Available Electronic Cigarettes in Mexico

**DOI:** 10.3390/ijerph21101308

**Published:** 2024-09-30

**Authors:** Alejandro Svarch-Pérez, María Vanessa Paz-González, Carlota Ruiz-Juárez, Juan C. Olvera-Chacón, Angelina Larios-Solís, Santiago Castro-Gaytán, Eugenia Aldeco-Pérez, Jorge Carlos Alcocer-Varela

**Affiliations:** 1Federal Commission for the Protection against Sanitary Risks, Oklahoma 14, Col. Nápoles, D.T. Benito Juárez, Mexico City 03810, Mexico; 2Center for Research and Technological Development in Electrochemistry, S.C., Parque Tecnológico Querétaro SN, Pedro Escobedo, Querétaro 76703, Mexico; mpaz@cideteq.mx (M.V.P.-G.);; 3Health Secretary, Homero 213, Piso 15 Col. Chapultepec Morales, D.T. Miguel Hidalgo, Mexico City 11570, Mexico; jorge.alcocer@salud.gob.mx

**Keywords:** e-cigarettes, e-liquids, aerosol

## Abstract

The chemical components of the e-liquids and aerosols contained in electronic nicotine delivery systems (ENDSs), better known as vapes, were evaluated. The analytical technique used was gas chromatography–mass spectrometry, where the extraction and injection methods were established in this study. The work consisted of the analysis of twenty samples of disposable electronic cigarettes prefilled with new e-liquid, of a known brand, flavor, volume, and, in some of them, the percentage of nicotine and the number of puffs per device were indicated on the label. We detected the presence of many substances (at a qualitative and semi-quantitative level), and we achieved the quantification of benzene, toluene, and xylenes (BTX), dangerous substances that cause severe damage to health. Several of the e-liquids and aerosols present BTX concentrations above the permissible exposure limit (PEL), recommended by the Occupational Safety and Health Administration (OSHA): benzene in aerosol samples 80% > PEL, and toluene in aerosol samples 45% > PEL. The number of chemical compounds found in the samples increases from 13 to 167, the average being 52 compounds for the water extraction method, 42 compounds for the methanol extraction method of e-liquids, and 107 compounds for the direct aerosol analysis. It is a fact that many of those compounds, especially BTX, can cause serious effects on human health, affecting the respiratory, digestive, cardiovascular, pulmonary, and immune systems, as well as the brain. Therefore, the use of these devices should be considered with caution, since the substances and their chemical nature may pose significant health risks to both users and those exposed to secondhand emissions.

## 1. Introduction

Nicotine, a potent psychoactive substance found in traditional cigarettes, is heavily regulated due to its high addictive potential and adverse health effects. Smoking poses significant public health risks, impacting both active and passive smokers, with severe damage primarily to the respiratory system and a range of serious diseases [1].

The concept of the so-called “smokeless cigarette” dates back to the 1960s, aiming to replicate the experience of smoking without emitting smoke, and in some cases, without using tobacco [2]. However, a significant breakthrough came in 2003 with the invention of the “atomizing electronic cigarette” by Hon Lik in China, which was later licensed and marketed by Ruyan starting in 2013 [3].

Initially, electronic cigarettes used nicotine extracted from natural tobacco and mixed with various fillers and additives to create the “e-liquids” or “electronic liquids”. These e-liquids are vaporized into an aerosol by heating, mimicking cigarette smoke. Promoted as an innovative smoking product, companies that manufacture ENDSs, have suggested that health risks could be minimized by delivering a mixture of nicotine and pleasant-smelling, flavorful chemicals through a modern device [4]. However, studies have indicated a potential link between vaping and lung injuries, known as EVALI (e-cigarette or vaping product-associated lung injury), particularly when additives like vitamin E acetate and THC (tetrahydrocannabinol) are involved [4,5,6,7].

Despite these risks, there is a lack of standardized labeling and regulation, particularly in countries like Mexico. Many e-liquid products do not provide adequate information about the nicotine content or other ingredients, raising concerns about potential health hazards, including addiction and respiratory damage due to unknown or harmful chemical compounds [8,9,10,11,12]. Common e-liquid components such as glycerin and propylene glycol, along with various additives, contribute to the complexity of the vaping process, which involves heating to 300 °C in a plastic container with an electrical current applied to a metallic filament [13,14,15,16,17].

Efforts to regulate vaping devices and e-liquids are ongoing in several countries, with restrictions being implemented [1,5,18]. In Mexico, for example, the General Law for Tobacco Control, article 16, section VI, effective from 2020, prohibits the marketing, distribution, and promotion of vaping devices [19,20,21,22,23].

Reports from the Federal Commission for the Protection against Sanitary Risks (COFEPRIS) have identified 33 chemical compounds in e-liquid samples, highlighting potential health risks but lacking quantitative data. As e-liquids become more prevalent, particularly with fourth-generation devices that allow easy modification of volume, flavor, and nicotine concentration, precise dosage regulation remains challenging [24,25].

Given the gaps in standardization and the emerging evidence of potential risks, it is crucial to develop methodologies for analyzing the content of the e-liquids and aerosols in disposable vapes. This study aims to provide a qualitative overview of the main components of some e-liquids available in Mexico and to propose a preliminary approach to quantifying hazardous substances such as BTX. With vaping still being relatively new and its risks not fully understood, especially regarding short-term use, this research is vital for addressing health and safety concerns [26].

## 2. Methodology

Karal brand methanol (HPLC-quality) and ultrapure water were purchased and used for the extractions and analysis. Individual stock extracts of e-liquids were obtained by 30-min sonication of the inner cartridges and stored in a closed glass vial at 4 °C.

Volatile organic compounds contained in the e-liquids and aerosols were detected and identified using GC-MS, Agilent Instrument, equipped with an EI source and a 7890A GC/7890B system, Agilent, detector MSD 5975C and 5977A (Agilent Technologies, Santa Clara, CA, USA). The library comparison data used was NIST08L. The ionization source and the quadrupole were maintained at 230 °C and 150 °C, respectively. Column types: DB-624 J&W Scientific 30 m × 320 mm × 1.8 μm and 2B-5MS plus 60 mm × 0.25 mm ID × 0.25 μm. The oven was maintained at 40 °C for 5 min, and then a gradient of 8 °C/min at 240 °C for 1 min was used. Helium was used as a carrier gas at a constant flow.

A batch of twenty new, disposable electronic cigarette devices were chosen to realize the analysis. Information such as brand, number of puffs, flavor, volume, and nicotine concentration are indicated on the label (Figure 1).

This batch was subjected to two types of analysis by gas chromatography coupled to mass spectrometry, one including extraction of the e-liquid, using water and methanol as solvents, and the purge and trap injection technique. The latter is achieved through thermal desorption of the aerosol generated by the vaping device, and both techniques were followed by mass-coupled gas chromatography.

The general methodology used in the e-liquid analysis is illustrated in Figure 2. Where the cartridge, whether liquid or absorbent material with the e-liquid, was extracted either with water or methanol, afterward the sample was subjected to sonication for 30 min and preserved at a temperature of 4 °C. For sampling, 500 microliters of sample were taken and diluted in organic-free water to 50 mL. The injection into the chromatography equipment was carried out through the purge and trap module at a temperature of 40 °C for 30 min.

The e-liquids were prepared for quantification with an internal standard, where the concentration values in both groups of samples (water and methanol extraction) of benzene, toluene, and total xylenes (BTX) were obtained.

In the case of aerosols, those were generated directly from each device (simulating vaping conditions) and were captured in a Tenax TA tube (MARKES), specially designed for the thermal desorption as an injection system and for the subsequent introduction to the chromatograph. This system allowed us to concentrate the aerosols through adsorption–desorption by temperature and gave us a general overview (semi-quantitative analysis) of the relative concentration in each sample, as well as quantitative analysis of the BTX.

Aerosols were sampled after five puffs, using two puffs of 2 s each and having 30 s between puffs, simulating two puffs of human use, see Figure 2. For more details in this section consult the Appendix A.

## 3. Results

The number of compounds identified through mass-coupled gas chromatography varied significantly, ranging from 13 to 167. This variation depends on the extraction and injection methods used, the phase of the sample, and the specific electronic cigarette device analyzed. Table 1 provides a detailed breakdown of the number of compounds detected in each electronic cigarette, along with the corresponding extraction and injection methods and the flavor profiles.

From the data presented in Table 1, it is evident that a greater number of compounds were detected in the aerosol phase when thermal desorption was used as the injection method, compared to the purge and trap method employed for extracting e-liquid. Previous studies have shown that new compounds are often formed during the vaping process, as e-liquid is exposed to temperatures exceeding 300 °C and transformed into aerosol [27,28]. These reactions, including decomposition, addition, and oxidation, increase both the diversity and number of compounds to which users are exposed. The oxidation of e-liquids, primarily glycerin, and propylene glycol, leads to the formation of harmful substances such as formaldehyde, acetaldehyde, acrolein, propanal, glyoxal, and methylglyoxal.

Figure 3 illustrates the GC chromatogram for sample 2, flavored Energy Ice, using the purge and trap method with water extraction performed on the e-liquid. In this sample, 71 organic volatile compounds were detected, including ethyl acetate (retention time, RT = 6.22), butanoic acid ethyl ester (RT = 12.07), *p*-xylene (RT = 13.89), beta-pinene (RT = 16.51), limonene (RT = 17.50), benzoic acid methyl ester (RT = 19.66), naphthalene (RT = 21.65), eugenol (RT = 24.91), vanillin (RT = 26.44), 2-methyl-1-propenyl-benzene (RT = 26.73), among others. Glycerin, propylene glycol, and nicotine were also detected as the most concentrated compounds. Chromatograms are included in the Appendix A.

The focus of the quantification analysis was on benzene, toluene, and total xylenes (BTX), as these compounds appeared frequently in the qualitative lists of components in the analyzed e-liquids. BTX are known to be highly toxic to humans through ingestion, exposure, and inhalation. Table 2 summarizes some of the health risks associated with BTX exposure. These compounds were quantified in both e-liquid and aerosol samples from electronic cigarettes.

Table 3, Table 4 and Table 5 display the quantification results for BTX in both e-liquids and aerosols. The presence of each analyte was confirmed by identifying the primary ion and employing an internal standard. For comparison, three parameters were used as reference points: (1) the mean concentration of BTX reported in combustible cigarettes by Pandey et al. [30], (2) the permissible exposure limit (PEL), meaning the maximum permitted 8-h concentration exposure, and (3) the short-term exposure limit (STEL), a 15-min time-weighted average exposure that should not be exceeded during a workday. These exposure limits are defined by the Occupational Safety and Health Administration in the Permissible Limits List for Chemical Contaminants [31].

The quantitative analysis of aerosols for BTX was conducted, with concentrations reported in ng/2 vapes. For comparison purposes, these values were converted to μg/L, considering the number of puffs and the volume declared for each device. Table 3 shows the minimum and maximum benzene concentrations in each extraction and GC injection method. For e-liquids extracted with water, 4 out of 20 samples had concentrations above the quantification limit, all of which were below the mean concentration found in cigarettes and within both PEL and STEL exposure limits. Methanol extraction yielded 12 out of 20 samples with concentrations above the quantification limit and below the cigarette concentrations and exposure limits. In aerosols, 12 out of 20 samples (60%) had benzene concentrations exceeding the mean cigarette concentration, 16 out of 20 (80%) exceeded the PEL, and 3 out of 20 (15%) were above the STEL. A list of all obtained BTX concentrations are included in the Appendix A. 

Table 4 presents the results for toluene. For e-liquids extracted with water, 9 out of 20 samples had concentrations above the quantification limit, all of which were below the mean concentration in cigarettes, PEL, and STELs. Methanol extraction yielded 13 out of 20 samples above the quantification limit, with all concentrations remaining below the reference limits. However, in aerosols, 11 out of 20 samples (55%) exceeded the mean cigarette concentration, 9 out of 20 (45%) exceeded the PEL, and none exceeded the STEL.

Table 5 shows the results for xylenes (total xylenes), where water extraction of the e-liquids yielded 15 out of 20 samples above the quantification limit, with 6 samples exceeding the mean cigarette concentration. Methanol extraction resulted in 16 out of 20 samples above the quantification limit, with 12 samples exceeding the mean cigarette concentration. In aerosols, 18 out of 20 samples (90%) had xylene concentrations above the mean cigarette concentration; with just one sample exceeding both the PEL and STELs (sample 7).

In electronic cigarettes, aerosols are formed directly at the mouthpiece outlet after the e-liquid comes into contact with the heating filament inside the device. The significance of analyzing the vapor lies in the fact that the compounds in the e-liquid mixture can transform into other substances due to temperature, reactions with other components, humidity, and/or air.

Consequently, comprehensive lists of the compounds present in the aerosols of twenty samples were generated, including both a semiquantitative list and a quantitative analysis of BTX, as presented in Table 3, Table 4 and Table 5. From the semiquantitative list, it was found that at least seven components were consistently present in most of the samples, with the highest estimated concentrations (based on toluene equivalents). These primary components of the aerosols include glycerin, benzoic acid, nicotine, propylene glycol, ethanol, butanoic acid ethyl ester, and ethyl acetate.

The secondary group of compounds includes aldehydes (both aliphatic and aromatic), which are often part of the flavor or result from decomposition reactions. Terpenes were also identified as components related to flavor and aroma.

## 4. Discussion

Twenty e-liquid and aerosol samples, from electronic cigarettes commercially available in Mexico, were analyzed by gas chromatography–mass spectrometry. Besides qualitative analysis, which gave us an average 67 number of compounds per sample, a quantitative study was achieved. Benzene, toluene, and xylenes were quantified in liquid and aerosol phases of the original e-liquid. Methodology and sample treatment is a self-developed procedure, using the purge and trap injection mode for the e-liquids and a thermal desorption module for the aerosols, coupled to the gas chromatograph and mass spectrometer.

A significant observation was that the aerosol phase consistently contained a greater number of compounds than the e-liquid phase across all samples. This may be attributed to two factors: the superior detection capability of the thermal desorption method, and/or the chemical transformations that occur during vaping, where molecules in the e-liquid are subjected to high temperatures, resulting in the formation of new compounds or decomposition into others. Notably, aldehydes were found in several samples, such as acrolein (for example in sample 3), a low molecular weight aldehyde known to be an irritant and a potential carcinogen due to its immunosuppressive effects [32,33,34]. Acrolein and/or some derivative of it (methacrolein, 2-propen-1-ol), is present in 14 out of 20 aerosol samples analyzed.

The quantification of BTX was a primary focus due to their known toxicity to humans, particularly through inhalation. The analysis of aerosols, which represents the actual mode of user exposure, revealed concerning results. In the case of benzene, while all concentrations in the e-liquid phase (both water and methanol extractions) were below the mean concentration found in cigarettes and within permissible exposure limits, the aerosol analysis told a different story. Sixty percent of aerosol samples exhibited benzene concentrations exceeding those found in cigarettes, 80% were above the permissible exposure limit (PEL), and 15% exceeded the short-term exposure limit (STEL), suggesting that the benzene concentration in aerosols poses a greater health risk compared to combustible cigarettes and regulatory limits [30,31].

For toluene, the e-liquid analysis indicated concentrations below the reference values, but in aerosols, 55% of the samples had concentrations above the mean found in cigarettes, with 45% exceeding the PEL. Although none of the aerosol samples surpassed the STEL for toluene, the highest recorded value was alarmingly close to this limit, underscoring the potential risks associated with vaping.

Xylenes presented a similar pattern. In e-liquids, 40% of the water-extracted samples and 75% of the methanol-extracted samples exceeded the mean cigarette concentration. In aerosols, 90% of the samples showed higher concentrations than those found in cigarettes, with one sample exceeding both the PEL and STELs.

It is important to note that the water extraction method for e-liquids is considered more representative than methanol, as it minimizes the interaction between the solvent and the e-liquid components, unlike methanol which may introduce a matrix effect, potentially altering the results.

While comparing e-liquid and aerosol concentrations must be conducted cautiously due to differences in sampling, extraction, and analysis methods, a clear trend emerges: aerosol phase concentrations are generally higher than those in the e-liquid phase. This finding is critical, as the BTX concentrations in aerosols, particularly benzene and toluene, frequently exceed both cigarette concentrations and regulatory limits, indicating a significant health risk.

In conclusion, while the qualitative and quantitative analysis of e-liquids is essential for regulation and quality control, the composition of aerosols provides crucial insights into the toxicity and health implications of electronic cigarette use, emphasizing the need for stringent regulatory oversight.

## 5. Conclusions

In this study, a batch of twenty new vaping devices, each with a known origin, and flavor was analyzed using a self-developed procedure. This methodology involved the use of the purge and trap injection mode for the e-liquids and a thermal desorption module for the aerosols, coupled to the gas chromatograph and mass spectrometer (GC-MS).

The methodology proved effective in providing valuable qualitative data on the components present in each e-liquid, as well as quantitative measurement of three key analytes with significant health implications: benzene, toluene, and total xylenes (BTX). For aerosols generated directly from the devices, the thermal desorption injection GC-MS approach allowed for the identification of a broad range of compounds, along with an estimated concentration per sample and the quantification of BTX.

The results from this study revealed that each e-liquid is a complex mixture containing between 13 and 71 compounds. In contrast, the aerosols were found to be even more complex, containing between 73 and 167 compounds—more than twice the number found in the e-liquids. Among these components, BTX were quantified in all aerosol and e-liquid samples. Notably, 80% of the aerosol samples exceeded the permissible exposure limit (PEL) for benzene, 45% exceeded the PEL for toluene, and one sample exceeded the PEL for xylenes.

Given the significant number of compounds detected, as well as the elevated levels of BTX, it is strongly recommended to exercise caution when considering the use of e-cigarettes as a healthier alternative to tobacco or for daily recreational use. The findings suggest that these devices may pose serious health risks, warranting further investigation and regulatory oversight.

## Figures and Tables

**Figure 1 ijerph-21-01308-f001:**
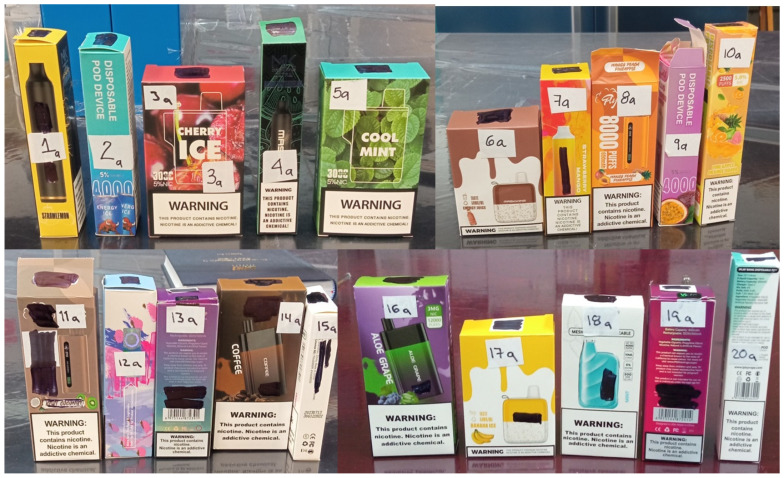
A batch of twenty disposable and new electronic cigarettes.

**Figure 2 ijerph-21-01308-f002:**
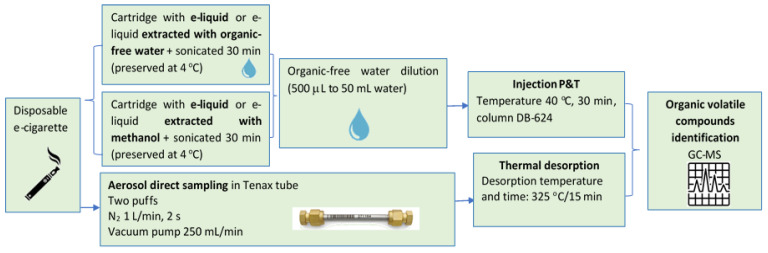
E-liquids and aerosol general methodology.

**Figure 3 ijerph-21-01308-f003:**
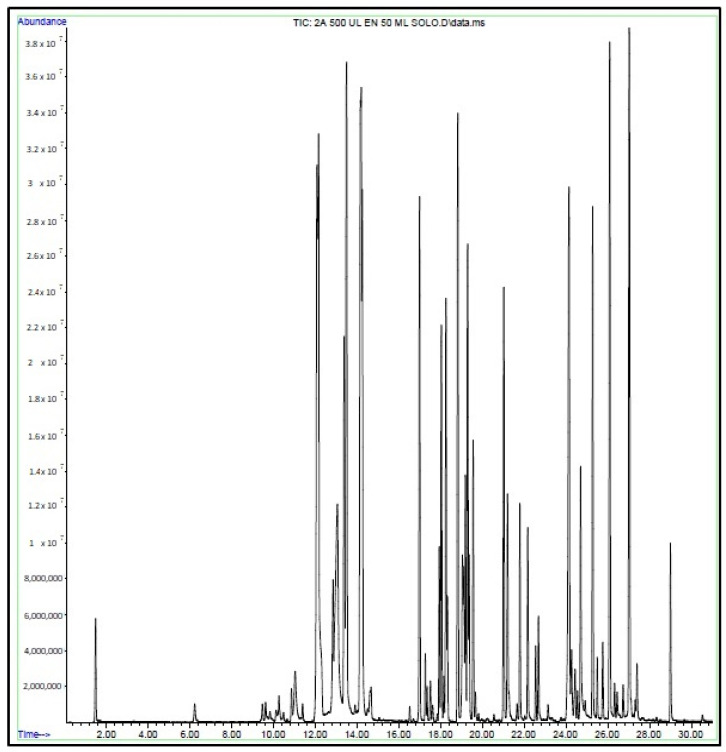
GC-MS chromatogram of sample 2. E-liquid, Energy Ice flavor, water extraction methodology, P&T injection.

**Table 1 ijerph-21-01308-t001:** Number of compounds detected in the e-liquids and aerosols.

	Number of Compounds Found			
Sample Number	E-Liquid Water Extraction, P&T	E-Liquid Methanol Extraction, P&T	Aerosol Thermal Desorption	Declared “Flavor”
1	53	47	130	Strawlemon
2	71	65	121	Energy ice
3	50	44	112	Cherry ice
4	61	58	120	Berry watermelon
5	56	41	108	Cool mint
6	63	61	100	Energy juice
7	54	52	112	Strawberry mango
8	57	51	101	Mango peach pineapple
9	54	54	115	Passion fruit
10	63	33	116	Pineapple orange guava
11	51	25	106	Kiwi coconut
12	55	57	122	Cherry cola
13	46	23	73	Blue razz ice
14	30	13	87	Coffee
15	55	32	167	Mamba
16	40	40	82	Aloe grape
17	39	37	90	Banana ice
18	51	45	104	Mint
19	49	28	87	Kiwi dragon berry
20	48	36	93	Cool mint
Average of components	52.3	42.1	107.3	-

**Table 2 ijerph-21-01308-t002:** BTX and induced human health damage [29].

Analyte	Human Health Damage
Benzene	Human carcinogen, affects the central nervous system and blood-forming organs
Toluene	Affects the central nervous system
Xylenes	Irritant, breathing difficulties, lung and liver damage

**Table 3 ijerph-21-01308-t003:** Benzene concentrations in e-liquids, aerosols, and reference parameters.

	Benzene				
Reference Concentration Parameters			Experimental Concentration Values		
Mean concentration found in cigarettes [30] g/L	Permissible exposure limit (PEL) * g/L	Short-term exposure limit (STEL) ^#^ g/L	E-liquid, in water extraction g/L	E-liquid in methanol extraction g/L	Aerosol estimated g/L
-		-	72 min	180 min	517 min
1295	1000	5000	680 max	817 max	9562 max

* PEL Maximum permitted 8-h time-weighted average concentration of an airborne contaminant [31]. # STEL A 15-min time-weighted average exposure which is not to be exceeded at any time during a workday [31]. min = minimum obtained value. max = maximum obtained value.

**Table 4 ijerph-21-01308-t004:** Toluene concentrations in e-liquids, aerosols, and reference parameters.

	Toluene				
	Reference Concentration Parameters		Experimental Concentration Values		
Mean concentration found in cigarettes [30] g/L	Permissible Exposure Limit (PEL) * g/L	Short-Term Exposure Limit (STEL) ^#^ g/L	E-liquid in water extraction g/L	E-liquid in methanol extraction g/L	Aerosol estimated g/L
-	-	-	88 min	278 min	115 min
7691	10,000	150,000	2562 max	4355 max	130,618max

* PEL Maximum permitted 8-h time-weighted average concentration of an airborne contaminant [31]. # STEL A 15-min time-weighted average exposure which is not to be exceeded at any time during a workday [31]. min = minimum obtained value. max = maximum obtained value.

**Table 5 ijerph-21-01308-t005:** Xylenes concentrations in e-liquids, aerosols, and reference parameters.

	Xylenes				
	Reference Concentration Parameters		Experimental Concentration Values		
Mean concentration found in cigarettes [30] g/L	Permissible exposure limit (PEL) * g/L	Short-term exposure limit (STEL) ^#^ g/L	E-liquid in water extraction g/L	E-liquid in methanol extraction g/L	Aerosol estimated g/L
-	-	-	163 min	778 min	799 min
1079 ^≠^	100,000	150,000	17,981 max	15,429 max	165,555 max

* PEL Maximum permitted 8-h time-weighted average concentration of an airborne contaminant [31]. # STEL A 15-min time-weighted average exposure which is not to be exceeded at any time during a workday [31]. ≠ m and *p*-xylenes were quantified. min = minimum obtained value. max = maximum obtained value.

## Data Availability

The data presented in this study are available on request from the corresponding author due to privacy of the information.

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
