# Peer review of "Methods for a Non-Targeted Qualitative Analysis and Quantification of Benzene, Toluene, and Xylenes by Gas Chromatography-Mass Spectrometry of E-Liquids and Aerosols in Commercially Available Electronic Cigarettes in Mexico"

_ijerph, 2024, doi:10.3390/ijerph21101308_

Round 1

Reviewer 1 Report

Comments and Suggestions for Authors

IJERPH – Manuscript ID  ijerph-3090453

Title: Quantification of benzene, toluene and xylenes by gas chromatography-mass spectrometry of e-liquids and aerosols in commercially available electronic cigarettes in Mexico.

Authors: Svarch-Pérez et al

Summary: The authors present methods for the analysis of e-liquids and electronic nicotine delivery systems (ENDS) aerosols. The method comprises a comprehensive, qualitative (and semi-quantitative) approach for the analysis of unknown components in disposable e-liquids and ENDS aerosols with reliance on the NIST08L library for compound identification as well as an included approach for the quantitative analysis of 3 specific analytes: benzene, toluene and xylenes. The authors utilized two e-liquid extraction methods, employing methanol and water as  extraction solvents in each method, as well as a thermal desorption method for the analysis of aerosols. The assessment by the authors identifies between 24 and 167 analytes depending on the specific device tested, and the method used. For the 3 quantitated analytes, the authors compare values found in e-liquids and aerosols to World Health Organization (WHO) ingestion limits and Public Health Service Agency for Toxic Substances and Diseases chronic inhalation limits to justify stronger regulation of disposable ENDS products in Mexico.

Reviewers Comments

Major Comments: 

1)      The title of the article does not accurately reflect the content of the article. The manuscript not only includes a quantitative approach for benzene, toluene and  xylene but the authors also describe  methods for a non-targeted, qualitative analysis and identification unknown compounds in  e-liquids and ENDS aerosols. The authors should consider revising the title of the manuscript to reflect the content..

2)      In the introduction the authors mention on Page 2, lines 50-52, the authors state “However, some research groups have found a relationship between vapes users and the development of lung injuries associated with the use of electronic cigarettes or vaping (EVALI).” The authors should correct this mischaracterization.  The EVALI reports were observed with illicit vaping products containing THC in vitamin E acetate and not with nicotine containing electronic cigarettes (Blount et al. Vitamin E Acetate in Bronchoalveolar-Lavage Fluid Associated with EVALI, N Engl J Med. 2020 Feb 20; 382(8): 697–705. The authors modify the statement to accurately reflect the facts.

3)      The authors should include a comparison of the benzene, toluene and xylenes relative combustible cigarettes.  Particularly since many of the users of these products are people who smoke cigarettes, the most hazardous tobacco product since the harm from cigarettes is caused by inhaling the smoke which include more than 7000 chemicals, 70 of which are carcinogens.  The harm reduction potential of electronic cigarettes must be included in the discussion.  Many authoritative bodies, including Public Health England (https://www.gov.uk/government/publications/nicotine-vaping-in-england-2022-evidence-update/nicotine-vaping-in-england-2022-evidence-update-summary#chapter-16-conclusions), have acknowledged that electronic cigarettes are less risky than cigarettes.

4)      The accuracy of the results cannot truly be assessed without a full description of the experimental details. While there is a brief discussion on the general approach to carrying out the experiment, it needs to be fully reproducible from the explanation of the method, even if it is only included in the supplemental information. Missing details which are critical include, but are not limited to, software used for deconvolution and deconvolution settings, approach to exclude artefactual peaks (blank subtraction, for example), criteria used to determine what constitutes a positive library match (some compounds presented in the supplemental information seem highly unlikely, while others appear to be related to column bleed). The accuracy of the results is dependent on sound methodology, and there are too few details presented herein. Additionally, the authors should include in the methods section, how quantitation was carried out, including internal standard used, standard curves and relevant statistical data.

5)      WHO determined ingestion limits are not applicable to e-liquids or ENDS aerosols. These are not ingested, and including these limits provides no value or clarity. Further, use of MRLs as an indicator of health implications is a misleading use of these values. The US EPA describes MRLs as below:

“MRLs are considered to be levels below which contaminants are unlikely to pose a health threat. Exposures above an MRL do not necessarily represent a threat, and MRLs are therefore not intended for use as predictors of adverse health effects or for setting cleanup levels.”

This is not to say that consideration of these toxic compounds is not important. However, as used in the article, the MRL of 0.003 ppm is considered the chronic MRL over the course of a year or more for benzene, and using this value is misleading to the reader. OSHA limits for safe exposure over the course of an 8 hour day is 1 ppm, for example, substantially higher than the MRL listed for chronic exposure. The authors should revise the manuscript to include comparisons to limits set by authoritative bodies by taking  into consideration the exposure based on use behavior of ENDS products (i.e., grazing over the day as opposed to acute, “bolus” dose)..

6)      Given that the authors report values for benzene, toluene and xylenes as  the key findings, please consider providing  levels for these analytes on a product-by-product basis (perhaps as a supplemental data file). This information will provide insights regarding the distribution of these analytes by product, beyond the minimum and maximum values reported. In addition, the authors should consider including a list of the compounds found in its entirety for each product in the supplemental data section, not just one single example product. Including such an expansive list in the article itself would indeed be cumbersome, but this may be of value and interest to the readers nonetheless.

.

7)      While analysis of e-liquids is important, in particular for product               quality assurance and adherence to any guidelines which may exist for various compounds in e-liquids, greater emphasis in the discussion should be made on the quantitated levels of compounds in aerosol, as this is the expected mode of exposure for the user of ENDS devices. What is important is how much of these analytes the user is exposed to when using the product as intended.

8)      Ethylbenzene secondary qualifier ions in the supplemental information do not appear to be aligned with the primary quantitation ion peak. This could be indicative of an incorrect compound identification. This should be examined further to confirm the correct identity, or otherwise explain why this misalignment is observed. The discussion as mentioned in point 1, indicating the importance of describing how library matches are assigned/confirmed, may help in this regard.

9)       The authors should consider including a limitations section in the discussion, to provide the reader with appropriate context of the findings reported in this manuscript. 

Author Response

Reviewer 1

Major Comments:

1)      The title of the article does not accurately reflect the content of the article. The manuscript not only includes a quantitative approach for benzene, toluene and xylene but the authors also describe  methods for a non-targeted, qualitative analysis and identification unknown compounds in  e-liquids and ENDS aerosols. The authors should consider revising the title of the manuscript to reflect the content.

Authors agree with the referee, we proposed a new title for the manuscript, so the new one could include the qualitative approach : “Methods for a non-targeted qualitative analysis and quantification of benzene, toluene, and xylenes by gas chromatography-mass spectrometry of e-liquids and aerosols in commercially available electronic cigarettes in Mexico”

2)      In the introduction the authors mention on Page 2, lines 50-52, the authors state “However, some research groups have found a relationship between vapes users and the development of lung injuries associated with the use of electronic cigarettes or vaping (EVALI).” The authors should correct this mischaracterization.  The EVALI reports were observed with illicit vaping products containing THC in vitamin E acetate and not with nicotine containing electronic cigarettes (Blount et al. Vitamin E Acetate in Bronchoalveolar-Lavage Fluid Associated with EVALI, N Engl J Med. 2020 Feb 20; 382(8): 697–705. The authors modify the statement to accurately reflect the facts.

The suggestion has been attended, text in original lines 50-52 were rewritten to “However, several research groups have found a relationship between vapes users and the development of lung injuries associated with the use of electronic cigarettes or vaping (EVALI) in some of the cases, where vitamin E acetate is present in addition to THC (tetrahydrocannabinol) as additives, according to some U.S. studies in a sample of 48 out of 51 patients. [4-7].” And the mentioned reference was also included as number 7. It is a fact that scientific evidence shows that just THC plus vitamin E acetate present in vapes, causes the EVALI, principally in open tank vapes or at the illicit market.

3)      The authors should include a comparison of the benzene, toluene and xylenes relative combustible cigarettes.  Particularly since many of the users of these products are people who smoke cigarettes, the most hazardous tobacco product since the harm from cigarettes is caused by inhaling the smoke which include more than 7000 chemicals, 70 of which are carcinogens.  The harm reduction potential of electronic cigarettes must be included in the discussion.  Many authoritative bodies, including Public Health England (https://www.gov.uk/government/publications/nicotine-vaping-in-england-2022-evidence-update/nicotine-vaping-in-england-2022-evidence-update-summary#chapter-16-conclusions), have acknowledged that electronic cigarettes are less risky than cigarettes.

Authors agree with reviewer about the comparison point, it will enhance the manuscript. Paragraph indicating the harm reduction potential is on lines 49-51, and indicated reference was included as [4].

Data about the concentration of BTX in cigarettes are now included in Tables 3, 4 and 5 and associated reference [30] Pandey, S.K.; Kim, K-H. Determination of Hazardous VOCs and Nicotine released from Mainstream Smoke by the Combination of the SPME and GC-MS Methods. The Scientific World Journal, 2010, 10, 1318-1329. DOI: 10.1100/tsw.2010.127.  Comparison between the danger level of combustible cigarettes and e-cigarettes must be careful taken since the consumer dose in both options is the important factor.

4)      The accuracy of the results cannot truly be assessed without a full description of the experimental details. While there is a brief discussion on the general approach to carrying out the experiment, it needs to be fully reproducible from the explanation of the method, even if it is only included in the supplemental information. Missing details which are critical include, but are not limited to, software used for deconvolution and deconvolution settings, approach to exclude artefactual peaks (blank subtraction, for example), criteria used to determine what constitutes a positive library match (some compounds presented in the supplemental information seem highly unlikely, while others appear to be related to column bleed). The accuracy of the results is dependent on sound methodology, and there are too few details presented herein. Additionally, the authors should include in the methods section, how quantitation was carried out, including internal standard used, standard curves and relevant statistical data.

Authors agree with the reviewer and the suggestion has been attended. Information about detailed methodology was now included in the supporting information section as no. 1 and 2.   Also a diagram of general methodology for aerosols are now included in Image 3 on manuscript. Data of the standard curves are in a PDF file, as part of the supporting information as well.

5)      WHO determined ingestion limits are not applicable to e-liquids or ENDS aerosols. These are not ingested, and including these limits provides no value or clarity. Further, use of MRLs as an indicator of health implications is a misleading use of these values. The US EPA describes MRLs as below:

“MRLs are considered to be levels below which contaminants are unlikely to pose a health threat. Exposures above an MRL do not necessarily represent a threat, and MRLs are therefore not intended for use as predictors of adverse health effects or for setting cleanup levels.”

This is not to say that consideration of these toxic compounds is not important. However, as used in the article, the MRL of 0.003 ppm is considered the chronic MRL over the course of a year or more for benzene, and using this value is misleading to the reader. OSHA limits for safe exposure over the course of an 8 hour day is 1 ppm, for example, substantially higher than the MRL listed for chronic exposure. The authors should revise the manuscript to include comparisons to limits set by authoritative bodies by taking  into consideration the exposure based on use behavior of ENDS products (i.e., grazing over the day as opposed to acute, “bolus” dose)..

Authors decide to change the comparison reference values from MRL to OSHA data, since it is true that the time factor on exposition is a crucial variable, more than ingestion. So, we decided better to add the information on Results section as Tables 3-5, where the reference values correspond now to the Permissible Exposure Limits (PEL), denoting to a maximum permitted 8 hour weighted average concentration of an airborne contaminant, and also the Short Term Exposure Limit, a 15 minute time weighted average exposure. Both values found in the U.S. Occupational Chemical Database, reference [31]. We then remove the information of ingestion limits and MRL’s, considering that OSHA parameters and data are closest to an aerosol’s environment exposition by inhalation comparison.

6)      Given that the authors report values for benzene, toluene and xylenes as the key findings, please consider providing levels for these analytes on a product-by-product basis (perhaps as a supplemental data file). This information will provide insights regarding the distribution of these analytes by product, beyond the minimum and maximum values reported. In addition, the authors should consider including a list of the compounds found in its entirety for each product in the supplemental data section, not just one single example product. Including such an expansive list in the article itself would indeed be cumbersome, but this may be of value and interest to the readers nonetheless.

Authors agree with reviewer and concentration data of all samples are now included in supplementary section as Table 1S, where values under the limits of quantification are noted.

Respect to the complete lists of compounds, they are now included as separated files, for water extracted samples, methanol extracted samples and also for the aerosol samples. They are in total 60 lists of compounds found.

7)      While analysis of e-liquids is important, in particular for product               quality assurance and adherence to any guidelines which may exist for various compounds in e-liquids, greater emphasis in the discussion should be made on the quantitated levels of compounds in aerosol, as this is the expected mode of exposure for the user of ENDS devices. What is important is how much of these analytes the user is exposed to when using the product as intended.

Comments of reviewer was attended. Aerosols are now in the discussion as the principal and most likely exposure mode, we agree that it is the more likely form that could imitate real consummation. Please find the related information in lines 143-145, 341-344.

8)      Ethylbenzene secondary qualifier ions in the supplemental information do not appear to be aligned with the primary quantitation ion peak. This could be indicative of an incorrect compound identification. This should be examined further to confirm the correct identity, or otherwise explain why this misalignment is observed. The discussion as mentioned in point 1, indicating the importance of describing how library matches are assigned/confirmed, may help in this regard.

The suggestion has been attended. About ethylbenzene, we observed a misaligning. To explain it, we wrote a note, included in the aerosol section of the supporting information, explaining the following:

“In the case of ethylbenzene probably present in the samples, in some cases the MS signal was not aligned with primary ion, and no coincidence on the abundance % was observed. This is due to an interference signal at the same retention time than ethylbenzene, sharing ions with the compound, increasing the concentration and the abundance %, changing the corresponding retention time. For this samples, a concentration below 10 ng (1st level curve) will be appropriate.”

9)       The authors should consider including a limitations section in the discussion, to provide the reader with appropriate context of the findings reported in this manuscript.

The suggestion was considered. Authors include the limitations in the discussion section, see lines 345-352, so that it will be clear the contribution of these work. Following statement was included in discussion section:

“Some limitations on this work are the following: Xylenes identification and quantification were done as total xylenes since it is though to distinguish each separate isomer. The e-liquid is a complex mixture, so limits somewhere the concentrations analysis and the GC equipment for some components found, for example for flavor molecules, HPCL would be a better analysis method instead. On the aerosol estimation for BTX concentration, we used the label volume and puffs for each device, nevertheless, the puff volume is not homogeneous for each device, since it depends on the physical characteristics of the e-cigarette, the tank or cartridge and the correct labeling of the product.“

Reviewer 2 Report

Comments and Suggestions for Authors

The manuscript entitled “Quantification of benzene, toluene and xylenes by gas chromatography-mass spectrometry of e-liquids and aerosols in commercially available electronic cigarettes in Mexico.” studied the detection of hazardous chemicals in electronic cigarettes using GC/MS.  It is a good idea and a well-written manuscript with updated references. The introduction has elements that integrate the work, allowing us to evaluate the context in which the manuscript is inserted. The methodology is suitable for its assessment but needs more explanation. The results are clear. The discussion is too short, more explanation should be linked to other studies. The conclusions are consistent with the evidence and address the main concept of the manuscript in detail. In general, the article represents a quality, however, minor comments are suggested below:

Comments:

1.     Lines 124-126, “Additionally, aerosols were generated directly from each device (simulating vaping conditions) and were captured in a Tenax TA tube, special for the thermal desorption as injection system and subsequent introduction to the chromatograph.” A diagram explains the aerosol sampling should be added.Please

2.     The discussion section is too short and more explanation should be added.

3.     There are no references in the discussion section. This section should discuss your results with other studies.

4.     The word “table” should be changed into “Table” at any place.

5.     The conclusion is too long. Please reduce. Also, add your recommendation.

6.     The references are appropriate.

7. I prefer to add the chromatograms in the manuscript.

8. In Image 2. E-liquids general methodology. Please correct inyection into injection. 

Author Response

Reviewer 2

Comments:

  1. Lines 124-126, “Additionally, aerosols were generated directly from each device (simulating vaping conditions) and were captured in a Tenax TA tube, special for the thermal desorption as injection system and subsequent introduction to the chromatograph.” A diagram explains the aerosol sampling should be added.Please

The suggestion has been attended, we added Image 3, including the aerosol sampling. Moreover, the procedure was also added in the supporting information.

  1. The discussion section is too short and more explanation should be added.

The suggestion has been attended, the discussion section has now references [30-33], comparison data and some more details.

  1. There are no references in the discussion section. This section should discuss your results with other studies.

Authors agree with reviewer, discussion is now longer and also including similar studies on combustible cigarettes and the OSHA permissible exposure limits for the chemical contaminants quantified. The discussion section has now references [30-33].

  1. The word “table” should be changed into “Table” at any place.

We thank for the remark, we already change the incorrected words.

  1. The conclusion is too long. Please reduce. Also, add your recommendation.

Authors agree with the suggestion, we have now a shorter conclusion section, more concise. We added a recommendation at the final sentence, lines 372-374  “According to the quantity of compounds observed but also the BTX quantification, we highly recommended to be very cautious with the acquisition and use of the e-cigarettes as a healthier alternative to tobacco or recreational daily use.”

  1. The references are appropriate.

Authors thank the reviewer for the comment, we added a couple or references.

  1. I prefer to add the chromatograms in the manuscript.

Authors attended the reviewer’s suggestion, we added Image 4 on manuscript, as an example, since we have in total 60 chromatograms. We will try to include them in the supporting information section.

  1. In Image 2. E-liquids general methodology. Please correct inyection into injection.

Thank you for the kind remark, Image 2 is now corrected.